# Relationship between Implant Geometry and Primary Stability in Different Bony Defects and Variant Bone Densities: An In Vitro Study

**DOI:** 10.3390/ma13194349

**Published:** 2020-09-30

**Authors:** Ahmad Ibrahim, Marius Heitzer, Anna Bock, Florian Peters, Stephan Christian Möhlhenrich, Frank Hölzle, Ali Modabber, Kristian Kniha

**Affiliations:** 1Department of Oral and Cranio-Maxillofacial Surgery, University Hospital RWTH Aachen, Pauwelstraße 30, 52074 Aachen, Germany; ahibrahim@ukaachen.de (A.I.); mheitzer@ukaachen.de (M.H.); abock@ukaachen.de (A.B.); flpeters@ukaachen.de (F.P.); fhoelzle@ukaachen.de (F.H.); amodabber@ukaachen.de (A.M.); 2Department of Orthodontics, University of Witten/Herdecke, Alfred-Herrhausen Str. 45, 58455 Witten, Germany; Stephan.Moehlhenrich@uni-wh.de

**Keywords:** dental implant, design, titanium, zirconia, ISQ, stability

## Abstract

Aim: This in vitro study aimed to evaluate the effects of implant designs on primary stability in different bone densities and bony defects. Methods: Five implant types (tapered-tissue-level, tissue-level, zirconia-tissue-level, bone-level, and BLX implants) were used in this assessment. The implants were inserted into four different artificial bone blocks representing varying bone-density groups: D1, D2, D3, and D4. Aside from the control group, three different types of defects were prepared. Using resonance frequency analysis and torque-in and -out values, the primary stability of each implant was evaluated. Results: With an increased defect size, all implant types presented reduced implant stability values measured by the implant stability quotient (ISQ) values. Loss of stability was the most pronounced around circular defects. Zirconia and bone-level implants showed the highest ISQ values, whereas tissue level titanium implants presented the lowest stability parameters. The implant insertion without any thread cut led to a small improvement in primary implant stability in all bone densities. Conclusions: Compared with implants with no peri-implant defects, the three-wall and one-wall defect usually did not provide significant loss of primary stability. A significant loss of stability should be expected when inserting implants into circular defects. Implants with a more aggressive thread distance could increase primary stability.

## 1. Introduction

The success or failure of dental implants, especially during the healing phase, is a multifactorial problem in which implant mobility/stability, according to the 1986 Albrektsson criteria is one of several key criteria [1]. One important prerequisite for achieving implant osseointegration is primary stability after implant placement [2]. Adequate initial stability is required to allow the implant to withstand micromovements before osseointegration is achieved. Subsequently, primary stability is one of several factors that is positively associated with secondary stability due to new bone formation [3]. Primary stability may be affected by many factors. Besides clinical factors such as resident stem cells, which can act as immunomodulatory and pro-osteogenic activities in the local environment, material-related factors such as implant design and drilling protocols play a key role, too [4,5].

On the one hand, drilling protocols, such as an underprepared drilling diameter [6,7] and the level of thread cutting at high density sites, affect primary implant stability. On the other hand, the implant’s micro- and macromorphologies are also related to the stability [8,9]. To date, many implant companies with many implant forms and concepts have been accumulated. Some factors are patient-related, such as bone density at the place of insertion, cortical bone layer thickness, or the incidence of bony defects [10].

Aside from implant design, implant material can also affect primary implant stability. Titanium implants are considered the gold standard for dental implantology, due to sufficient long-term data [11]. Conversely, zirconia implants are considered an alternative implant material because of their comparable osseointegration potential and natural teeth color matching [12,13,14]. However, because of the novelty of zirconia, there is currently a lack of valuable long-term data in the literature.

How can primary stability be checked in individual cases? Aside from insertion torque, a non-invasive conventional clinical mobility testing also gives an indication of osseointegration and bone–implant contact, due to the subjectivity and reliability of this method. In 1998, a resonance frequency analysis (RFA) was introduced as a non-invasive, objective, and reliable clinical technique [15]. The bone–implant contact (BIC) interface is detected from the RFA by a reaction of oscillations to the implant–bone contact. The measuring unit of RFA is defined as the implant stability quotient (ISQ). The values range from 1 to 100, with good stability being indicated by a high ISQ value (>60) and vice versa.

Many factors can influence the ISQ values, such as implant height, shape, stiffness, surface treatment, defect, healing time, and marginal bone loss. Bone defects around dental implants manifest in various forms and affect primary stability. They can be classified according to the Goldman and Cohen classification into one, two, three, or combined wall defects around the teeth and implants [16]. Especially in the case of immediate implantation or simultaneous implantation with bone augmentation, increasing peri-implant bone defects occur, which may affect the individual implant primary stability. Therefore, this study aimed to evaluate the effects of implant design and material on primary stability in terms of different bone densities and bony defects with comparable implant dimensions. The authors hypothesized that a more aggressive thread design leads to increased implant stability. Additionally, the influence of thread cutting on stability was investigated.

## 2. Material and Methods

### 2.1. Experimental Protocol

Five implant types, namely a tapered-tissue-level (8 mm length and diameter 4.1 mm), a tissue-level (8 mm length and diameter 4.1 mm), a zirconia-implant-tissue-level (8 mm length and diameter 4.1 mm), a bone-level (8 mm length and diameter 4.1 mm), and a BLX implant (8 mm length and diameter 4.0 mm), were used in this assessment (Institute Straumann AG, Basel, Switzerland, Figure 1).

Prior to implant placement, the examiner was calibrated by placing 10 additional implants for each design, according to the manufacturer’s instructions. Based on the manufacturer’s instructions for each implant type, a conventional drilling procedure, using a complete drill sequence (with individual surgical pilot-, twist-, and profile drills of 2.2, 2.8, and 3.5 mm in diameter), was adopted to prepare implant beds of 8 mm deep. After each implant drill, the depth was measured by using an implant-depth gauge (Institute Straumann AG, Basel, Switzerland). Except for the BLX implants with a self-cutting thread design, all implant types were inserted one time with and without thread cutting.

The implant site was prepared in four different artificial bone blocks (#1522-01, #1522-03, #1522-04, and #1522-05; Sawbones, Malmö, Sweden). This material has been approved by the American Society for Testing and Materials and has been recognized as a standard for testing orthopedic devices and instruments, making it ideal for the comparative testing of dental implants (American Society for Testing and Materials F-1839-08). These polyurethane foam blocks were classified into four different bone density groups: D1 (very dense bone), D2 (dense bone), D3 (porous bone), and D4 (very porous bone) [17].

The defect preparation was carried out according to the previously published classification and procedure of Shin et al. [18]. Three types of defects were prepared by using cylindrical and trepan drill burs (three-wall defect, one-wall defect, and circumferential defect, Figure 2). An implant site without any defect was used as the control group. Each defect was prepared at a 4 mm depth, and the defect size was controlled by using a standard dental probe. Furthermore, the centers of the implant bed were kept at a distance of at least 15 mm apart from each other.

A total of 1440 implants were inserted into the artificial bone blocks with different densities, using a repetition rate of 10 times in each subgroup. Every implant was inserted 10 times and then replaced with a new one. During implant placement with a special implant drive unit (Implantmed, W&H, Bürmoos, Austria), the torque-in value was recorded. To avoid any material damage, the maximum torque-in and -out values were set to 50 Ncm. Using resonance frequency analysis with hand-screwed individual smart pegs (Osstell, Gothenburg, Sweden), primary stability was evaluated after the insertion of the implant. Primary stability was measured with the ISQ value in four directions (i.e., from left and right and from front and back [18]), resulting in a calculated mean ISQ value. In the subsequent explanation, the torque-out value was measured with the same implant drive unit.

### 2.2. Statistical Analysis

The sample size was calculated by using G*Power software (G*Power, Version 3.1.9.2, Düsseldorf, Germany, Faul et al. [19,20]). The a-priori test (Wilcoxon–Mann–Whitney test for two groups) was used as an indication. The authors hypothesized that a more aggressive thread design leads to increased implant stability. Using a 0.05 significant level, an effect size of 5.16 (mean 1:72.6; standard deviation 1:2.4; mean 2:61.2; standard deviation 2:2 [21]) and power of 80%, at least *n* = 6 implants per group would be needed to verify the hypotheses.

Analyses were performed by using Prism 8 software for Mac OS X (GraphPad, La Jolla, CA, USA), running on Apple OS X. Before analysis, the values were tested for normal distribution, using the Shapiro–Wilk normality test. The groups were analyzed by using two-way ANOVA with the Geisser–Greenhouse correction. A post hoc Tukey’s multiple comparison test, with individual variances computed for each comparison, was conducted. We assessed any effect in the statistical model as significant if the corresponding *p*-value was below the 5% margin. Spearman’s rho test was applied, to evaluate the correlations between the ISQ and torque-in values. The values were considered “very weak” (0.00–0.19), “weak” (0.20–0.39), “moderate” (0.40–0.59), “strong” (0.60–0.79), or “very strong” (0.80–1.0) [22].

## 3. Results

The evaluation showed that, with an increased defect size, all implant types presented reduced implant stability values as measured by the ISQ values for all bone qualities from D1 to D4 (Figure 3 and Figure 4). Loss of stability was most pronounced in the circular defect (Table 1). This difference between no defects and circular defects was significant in all cases (*p* < 0.0500, Table 2). Implants placed in three-wall defects and one-wall defects also showed significantly better implant stability compared with circular defects, except for bone-level implants in D1 bone (*p* = 0.1916) and tissue-level implants in D4 bone (*p* = 0.0548). With a few exceptions (statistical difference), such as the comparison between no defect and three-wall defect around bone-level implants in D4 bone (*p* = 0.0091), the results between the no defect, three-wall, and one-wall defect groups, for the most part, did not provide a significant difference.

The stability measurement of the tapered implants with and without thread cutting revealed that this design delivered reduced ISQ values in D1 and D2 bone densities; however, in softer D3 and D4 bone qualities, tapered implants showed increased ISQ values compared with all other implant types. In our comparison, the lowest stability values were measured around tissue-level titanium implants, particularly in softer bone structures. By contrast, zirconia- and bone-level implants showed the highest ISQ values in various bone comparisons. The results indicated a significant difference between tissue level and zirconia implants in all defect and bone-density subgroups with thread cutting (*p* < 0.0500, Table 2). The comparison between the subgroups with and without thread cut showed no major differences, with the standard deviations being larger without thread cutting. Regarding the subgroup without thread cutting, BLX implants showed significantly lower stability values than zirconia and tapered implants, especially in soft D4 bone (*p* < 0.0500, Table 2).

Overall, the torque-in values were higher than the torque-out values in all subgroups (Table 3). Thread cutting led to a reduction of the torque-in and torque-out values when compared to those without thread cutting. The omission of the thread cutting led to a slight increase in ISQ values in all bone densities. The ISQ values ranged from the ISQ maximum of 60.1 (without thread cut 56.8) in D1 bone to the ISQ minimum of 29.4 (without thread cut 27.3) in D4 bone.

A very strong significant correlation between the ISQ and torque-in values was measured for either implants with thread cutting (Spearman’s rho 0.8655, *p* < 0.0001) or without thread cutting (Spearman’s rho 0.8155, *p* < 0.0001). The torque-in values ranged from the ISQ maximum 33.7 (without thread cut 41.0) in D1 bone to the ISQ minimum of 4.2 (without thread cut 5.4) in D4 bone.

## 4. Discussion

Misch et al. classified bone quality into four types (D1–D4) [17]. Primary stability is usually lower in type D4 bone than in types D1–D3 [23]. Arosio et al. showed that an increase in the diameter of the implant increased implant stability more than an equal increase in the implant length [10]. Conversely, the current in vitro study focused on different implant body- and thread designs with comparable sizes and different peri-implant defects. The aim was to evaluate the effect of the implant design and material on primary stability, in terms of different bone densities. In an ex vivo peri-implant bovine rib bone model, Shin et al. found that the defect size increased, while the ISQ values decreased [18]. Similarly, our results showed that, with an increased defect size, all implant types had reduced implant stability values, but loss of stability was most pronounced in the circular defects. This finding agrees with those of two studies, which found that implant stability showed a significant drop when 50% of the implant surface was not embedded in the bone [24,25]. This observation indicates that primary stability is hampered above a certain threshold of peri-implant bone defect.

Threads contribute to primary stability by increasing the initial contact with the underlying bone [26]. Self-tapping implants are usually designed to avoid the use of thread-cutting techniques to prepare the implant site, which is replaced by the action of cutting edges integrated into the lower apical part of the implant. Threads, however, differ in the way they transmit loads to adjacent bone [26]. V-shaped threads and implants with reverse buttress-threaded implants showed to transmit axial force through a combination of compressive, tensile, and shear forces. Furthermore, as the thread face angle increases, the shear forces generated by the various thread shapes increase, too [27]. Non-self-cutting blades create a lateral compression with an increased contact surface area, thus improving primary stability. In our study, the non-self-cutting threads showed higher primary stability than the self-cutting threads. These findings were in agreement with the results of Kim et al. regarding the self-cutting BLX implants in D4 bone but not in D1 bone, where BLX implants also showed higher ISQ values [28].

Moreover, compared with dental implants with parallel walls, the tapered shape may lead to better primary stability due to an improved compressive force distribution. However, several studies have reported controversial results with lower stability in tapered implants than in parallel ones [29,30], whereas other studies have found the opposite [31,32]. In our study, the tapered effect might have positively influenced implant stability especially in D4 bone density and circular effect due to the improved compressive forces at the implant tip.

The comparison between implant stability measurements, such as torque-in and torque-out values, resonance frequency analysis, and percussion energy response, remains controversial [33]. The torque out values were smaller when compared to the insertion torque. This is due to the correlation of the torque out value with the gripping volume, whereas the insertion torque value is rather associated with individual bone compression, e.g., due to different bone densities [34]. Resonance frequency analysis and torque-in and -out values may provide an objective approach to measure initial primary stability. Resonance frequency analysis can detect changes in micromotion that may be associated with an increase or decrease in the degree of stability. For this reason, resonance frequency analysis has been extensively used in in vitro studies, to compare different implant designs [33,35,36,37]. However, one author argued that different implant types should not be compared and that resonant frequency analysis should be used for follow-up of the same implant [38,39]. Owing to this current disagreement in the literature, we recommend that comparisons of different implant types be made with caution. Furthermore, it should be noted that BLX implants have a 0.1 mm smaller diameter than all other types in this study.

After this initial in vitro experiment presented here, clinical healing processes should also be considered. For example, the clinical microenvironment of the wound is a key factor in the implant integration. In regenerative medicine dental mesenchymal stem cells are important in terms of their potential to differentiate into osteogenic lines around implants [4,5]. They may have the future potential to improve bone healing, especially in relation to peri-implant bony defects. Furthermore, biomaterials such as Platelet-Rich Fibrin could be used to fill the defects around implants between bone tissue and the implant surface in a clinical scenario [40].

Furthermore, Bardyn et al. evaluated the lack of consistency between the presented methods (resonance frequency analysis and torque-in and -out values) and showed no common protocol for primary stability assessment [41]. Our findings disagree with this, as we found a very strong significant correlation between ISQ and torque-in values.

Although one calibrated surgeon placed all implants by using standardized drilling protocols, a fully guided implant bed preparation has been shown to be associated with a higher primary stability [33]. The drill hole should be more exact with undersized drilling, using fully guided surgery than with standard drilling, resulting in improved values of primary stability. A critical reflection on this study is that no fully guided implant placement was performed and that no separated cortical and cancellous structures were simulated, which could have affected the accuracy of the stability measures. The missing cortical structure could also explain why the maximum ISQ value (range 1–100) only reached 56.8 in our study.

## 5. Conclusions

In terms of primary stability, our results indicated that no significant loss of stability should be expected in smaller three-wall and one-wall peri-implant defects. However, a significant reduction in primary stability was observed especially in circular defects. As a counteract, a slight increase in primary stability could be achieved by omitting the thread cut bevor the implant placement.

## Figures and Tables

**Figure 1 materials-13-04349-f001:**
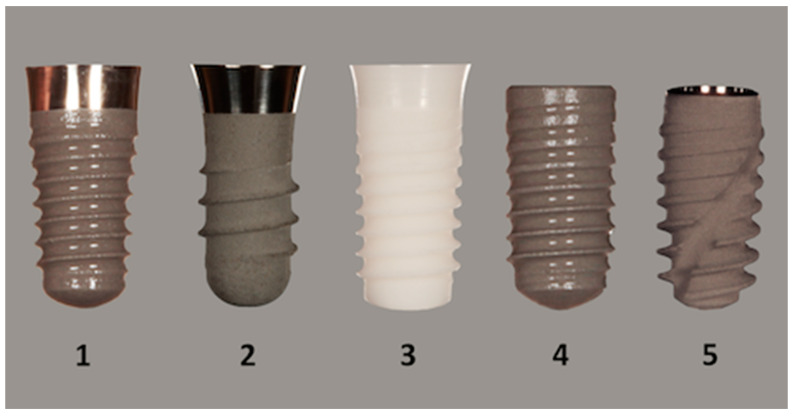
All five implant types, with a length of 8 mm, used in this investigation. Position 1 shows the tapered tissue level, position 2 shows the tissue level, position 3 shows the zirconia implant, position 4 shows the bone level, and position 5 shows the BLX implant (Institut Straumann AG, Basel, Switzerland).

**Figure 2 materials-13-04349-f002:**
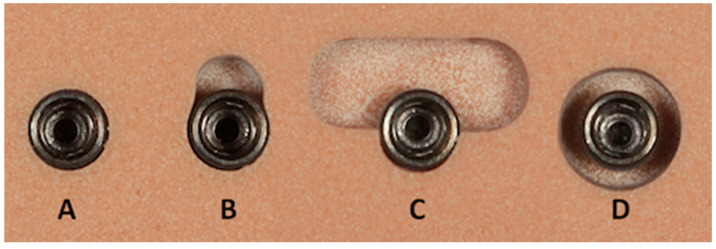
The control ((**A**) no defect) and three different types of defects were prepared ((**B**) three-wall defect, (**C**) one-wall defect, and (**D**) circumferential defect). Each defect was set at a depth of exactly 4 mm.

**Figure 3 materials-13-04349-f003:**
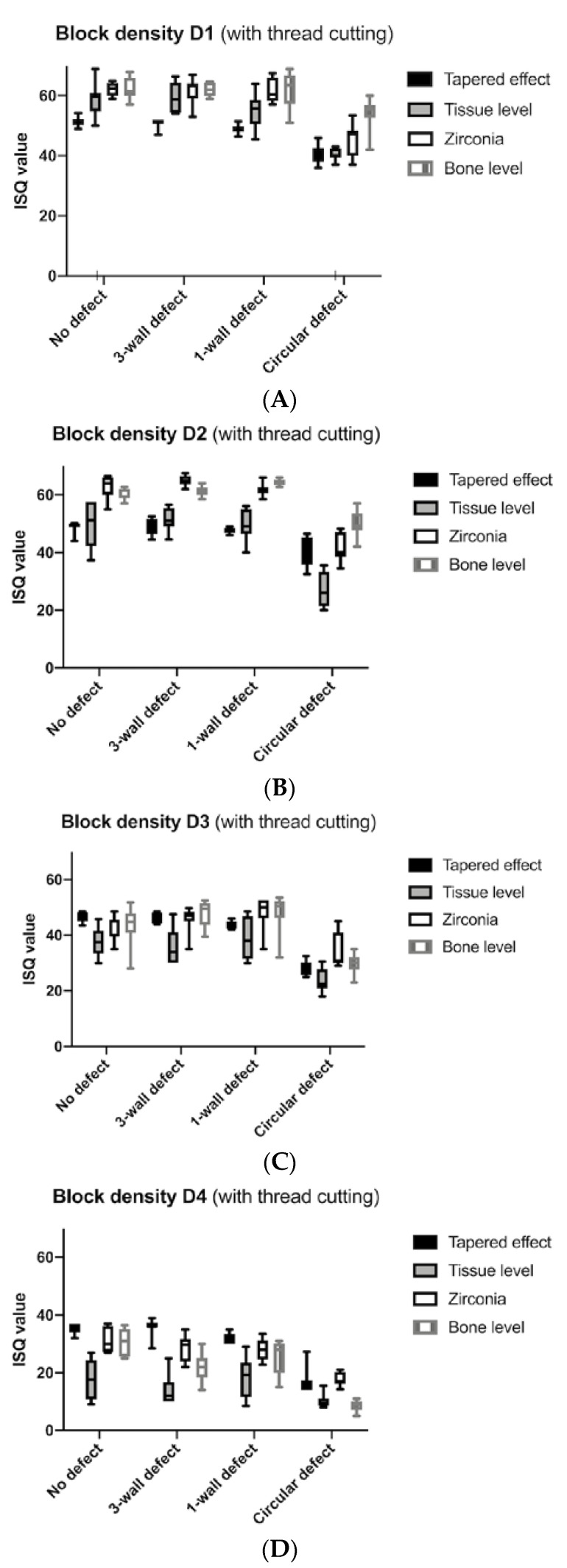
(**A**–**D**) implant stability quotient (ISQ) values of the four implant types that were inserted with thread cutting in D1, D2, D3, and D4 bone densities.

**Figure 4 materials-13-04349-f004:**
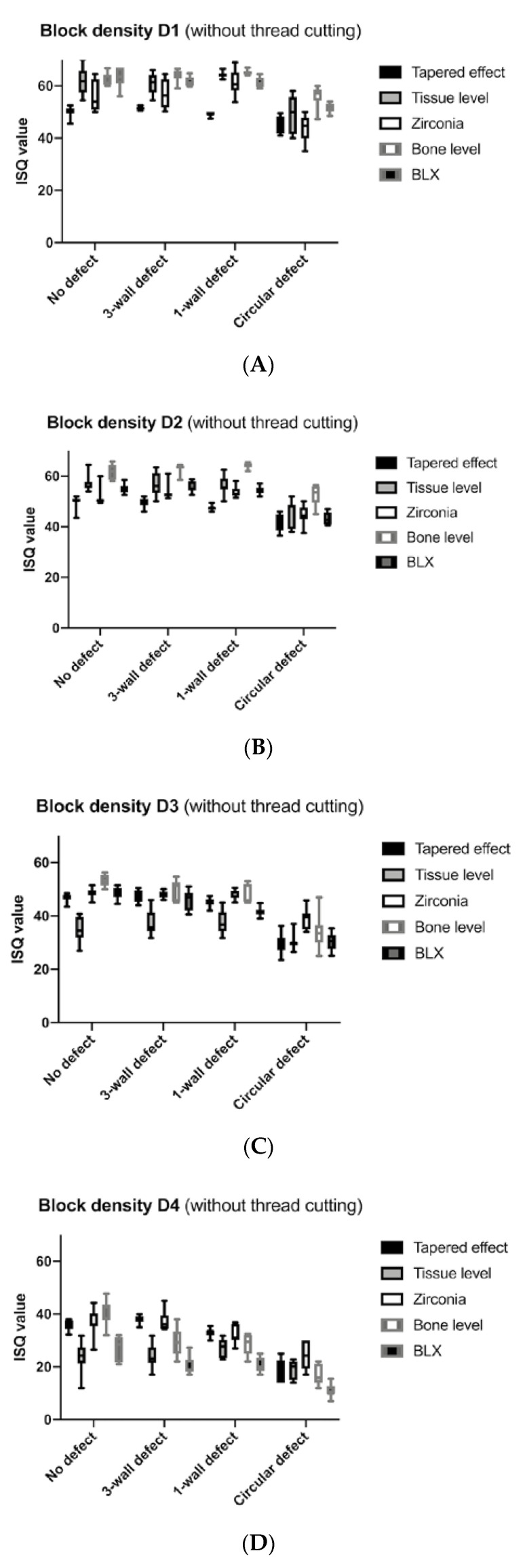
(**A–D**) ISQ values of the five implant types (including the BLX) that were inserted without thread cutting in the D1, D2, D3, and D4 bone blocks.

**Table 1 materials-13-04349-t001:** Descriptive parameters of the ISQ and torque-in and -out values.

	Type of Implant	Defect Size	With Thread Cutting			Without Thread Cutting		
		Block Density	ISQ Mean (SD)	Max Torque in Value (SD)	Max Torque out Value (SD)	ISQ Mean (SD)	Max Torque in Value (SD)	Max Torque out Value (SD)
D1	Tapered effect	no defect	51.3 (1.5)	49.9 (0.3)	38.6 (6.3)	50.0 (0.0)	50.0 (0.0)	37.5 (3.7)
		3-wall defect	51.0 (1.5)	43.9 (7.4)	30.9 (4.9)	51.4 (0.6)	47.5 (3.3)	33.2 (1.9)
		1-wall defect	49.0 (1.5)	47.2 (3.3)	33.9 (2.0)	49.2 (0.8)	50.0 (0.3)	35.2 (1.9)
		Circular defect	40.6 (3.0)	41.4 (4.9)	26.7 (2.3)	45.4 (3.1)	47.3 (2.5)	31.9 (2.6)
	Tissue level	no defect	58.9 (5.1)	15.3 (5.3)	12.5 (3.4)	61.9 (5.0)	35.4 (9.0)	27.1 (5.2)
		3-wall defect	59.2 (5.0)	24.5 (9.2)	13.1 (5.1)	60.9 (3.9)	35.5 (6.9)	21.6 (2.3)
		1-wall defect	55.4 (5.5)	21.5 (5.4)	10.3 (2.8)	64.2 (1.1)	37.2 (5.7)	22.8 (4.7)
		Circular defect	40.6 (1.9)	16.9 (3.0)	6.2 (1.8)	49.5 (6.9)	25.4 (6.0)	12.4 (2.8)
	Zirconia	no defect	62.3 (2.2)	26.9 (4.6)	24.7 (4.3)	56.0 (5.8)	50.0 (0.0)	38.5 (3.4)
		3-wall defect	61.6 (4.1)	40.0 (4.7)	29.2 (1.6)	56.8 (5.5)	48.7 (2.4)	32.6 (3.4)
		1-wall defect	61.6 (3.8)	47.6 (3.4)	28.0 (4.2)	61.2 (4.6)	49.7 (0.9)	31.4 (2.6)
		Circular defect	45.6 (5.1)	29.3 (5.8)	14.8 (2.8)	44.1 (4.7)	32.8 (5.7)	17.7 (3.3)
	Bone level	no defect	62.3 (3.4)	41.6 (7.2)	26.3 (8,5)	62.0 (2.3)	50.0 (0.0)	37.1 (2.2)
		3-wall defect	62.0 (2.1)	24.5 (5.0)	21.0 (5.2)	63.9 (2.2)	38.9 (3.1)	31.2 (2.6)
		1-wall defect	62.1 (6.0)	38.1 (10.3)	27.1 (9.8)	65.1 (0.8)	47.5 (3.6)	37.1 (3.1)
		Circular defect	54.1 (5.0)	28.6 (4.4)	20.8 (3.3)	56.1 (3.9)	33.3 (7.6)	22.7 (2.7)
	BLX	no defect				63.2 (3.8)	40.5 (3.9)	26.0 (2.4)
		3-wall defect				61.7 (1.9)	33.4 (3.9)	19.9 (2.8)
		1-wall defect				61.3 (1.8)	39.2 (5.2)	25.5 (2.1)
		Circular defect				51.4 (1.8)	28.3 (5.8)	17.8 (8.6)
D2	Tapered effect	no defect	49.0 (1.9)	29.2 (4.1)	27.4 (3.7)	49.9 (2.3)	39.8 (5.7)	30.5 (5.9)
		3-wall defect	49.5 (3.0)	41.5 (5.6)	26.2 (3.6)	49.7 (1.9)	46.5 (3.7)	29.4 (2.1)
		1-wall defect	47.7 (1.1)	39.0 (3.7)	27.4 (2.8)	47.4 (1.0)	49.7 (0.7)	31.5 (2.3)
		Circular defect	41.1 (4.9)	30.8 (2.3)	18.0 (1.9)	41.3 (3.4)	36.7 (2.3)	24.1 (2.8)
	Tissue level	no defect	49.5 (8.3)	8.3 (4.1)	4.8 (3.3)	56.8 (3.0)	21.7 (5.5)	14.8 (2.4)
		3-wall defect	51.7 (3.9)	16.4 (4.3)	7.8 (2.2)	56.7 (4.3)	27.8 (6.4)	14.2 (1.9)
		1-wall defect	49.4 (5.0)	31.3 (2.1)	10.9 (2.1)	56.2 (3.4)	40.7 (4.4)	17.7 (2.8)
		Circular defect	27.2 (5.7)	12.3 (1.6)	2.3 (0.9)	45.8 (5.2)	26.1 (3.3)	8.8 (1.6)
	Zirconia	no defect	62.9 (3.9)	27.8 (3.0)	21.7 (3.6)	51.2 (3.1)	39.8 (4.8)	30.7 (4.3)
		3-wall defect	64.9 (1.8)	30.7 (3.5)	21.0 (2.1)	53.3 (2.8)	38.8 (2.1)	28.5 (2.0)
		1-wall defect	61.6 (2.0)	35.1 (4.4)	23.9 (4.2)	53.7 (1.9)	45.5 (4.4)	29.7 (2.5)
		Circular defect	41.4 (4.6)	21.7 (6.2)	10.7 (2.9)	44.8 (3.6)	28.5 (5.8)	17.1 (2.3)
	Bone level	no defect	60.7 (1.9)	24.9 (6.5)	20.8 (6.8)	61.6 (3.0)	49.2 (2.5)	39.7 (6.4)
		3-wall defect	61.3 (1.6)	23.1 (3.0)	17.4 (3.0)	63.2 (1.7)	32.6 (1.7)	25.6 (1.9)
		1-wall defect	64.3 (1.1)	32.5 (4.0)	26.2 (2.9)	64.2 (1.2)	43.4 (3.8)	31.0 (2.3)
		Circular defect	50.0 (4.2)	27.5 (6.4)	17.0 (4.8)	52.5 (4.3)	28.4 (4.6)	18.3 (3.4)
	BLX	no defect				55.2 (2.0)	24.1 (3.1)	15.1 (1.7)
		3-wall defect				56.5 (2.3)	25.2 (3.2)	15.8 (2.0)
		1-wall defect				54.3 (1.6)	19.4 (2.5)	13.6 (3.0)
		Circular defect				43.4 (2.4)	15.2 (2.4)	8.6 (0.7)
D3	Tapered effect	no defect	46.3 (1.7)	12.7 (1.1)	8.8 (0.6)	46.9 (1.6)	22.7 (1.5)	16.0 (1.3)
		3-wall defect	45.9 (1.8)	21.3 (3.5)	16.0 (2.0)	47.6 (2.2)	24.3 (2.6)	18.4 (2.8)
		1-wall defect	43.8 (1.4)	20.4 (1.4)	13.1 (1.5)	45.1 (1.6)	25.2 (2.8)	17.8 (1.8)
		Circular defect	28.5 (2.5)	8.1 (0.9)	4.8 (1.0)	29.4 (3.8)	15.7 (2.0)	9.5 (1.7)
	Tissue level	no defect	37.4 (5.3)	9.7 (3.7)	4.0 (2.6)	35.0 (4.5)	7.1 (1.0)	4.2 (1.6)
		3-wall defect	35.6 (6.2)	8.8 (4.6)	2.1 (2.5)	37.4 (4.6)	7.7 (0.7)	5.2 (0.8)
		1-wall defect	38.9 (7.0)	10.2 (1.0)	1.3 (0.9)	37.6 (4.6)	16.3 (3.0)	5.6 (0.7)
		Circular defect	24.0 (4.3)	7.1 (1.0)	0.9 (0.7)	30.2 (2.7)	8.6 (0.8)	3.0 (1.6)
	Zirconia	no defect	41.5 (4.2)	14.3 (2.2)	10.2 (1.8)	48.5 (2.1)	22.4 (1.6)	17.2 (0.9)
		3-wall defect	46.0 (4.1)	15.2 (3.6)	10.9 (2.7)	47.9 (1.2)	23.7 (2.3)	16.5 (1.6)
		1-wall defect	48.3 (5.4)	14.1 (1.4)	10.4 (1.3)	47.7 (1.7)	18.7 (1.3)	15.9 (1.4)
		Circular defect	33.8 (6.0)	9.1 (1.9)	5.8 (0.6)	38.9 (3.9)	12.9 (1.5)	8.0 (1.2)
	Bone level	no defect	43.7 (6.9)	9.3 (2.5)	4.3 (2.9)	53.3 (2.0)	15.7 (2.0)	11.5 (1.0)
		3-wall defect	47.6 (4.8)	8.4 (2.6)	5.1 (2.7)	49.9 (3.8)	13.5 (1.8)	9.4 (1.4)
		1-wall defect	48.1 (6.9)	13.1 (2.0)	8.6 (2.0)	49.5 (3.2)	16.8 (1.8)	11.5 (1.4)
		Circular defect	29.7 (3.4)	7.8 (1.0)	4.7 (1.3)	34.0 (5.9)	9.1 (1.7)	5.5 (1.4)
	BLX	no defect				48.5 (2.1)	10.6 (1.2)	7.0 (0.7)
		3-wall defect				46.0 (3.5)	8.7 (1.4)	5.6 (1.0)
		1-wall defect				41.4 (1.7)	7.5 (0.5)	6.7 (0.9)
		Circular defect				30.3 (3.4)	6.5 (0.7)	4.1 (1.0)
D4	Tapered effect	no defect	35.5 (1.7)	7.0 (0.7)	5.5 (0.7)	36.2 (2.0)	11.2 (1.4)	7.7 (1.1)
		3-wall defect	35.8 (2.8)	6.7 (0.5)	3.4 (1.2)	38.1 (1.5)	8.8 (0.9)	5.3 (0.9)
		1-wall defect	31.8 (1.7)	7.9 (1.0)	5.2 (0.9)	32.9 (1.5)	8.4 (1.1)	6.3 (0.8)
		Circular defect	16.8 (4.0)	6.4 (0.8)	4.3 (1.2)	18.7 (4.2)	7.5 (1.3)	4.9 (1.1)
	Tissue level	no defect	17.8 (6.8)	1.0 (2.1)	0.1 (0.3)	23.7 (5.7)	0.5 (1.0)	0.6 (1.0)
		3-wall defect	14.0 (5.4)	2.0 (2.3)	0.1 (0.3)	24.2 (4.3)	4.7 (2.1)	0.4 (0.5)
		1-wall defect	18.2 (6.8)	1.8 (2.4)	0.3 (0.9)	27.3 (3.4)	5.2 (1.0)	0.4 (0.5)
		Circular defect	10.3 (2.6)	3.2 (1.5)	0.4 (0.5)	19.1 (3.4)	4.9 (2.0)	0.4 (0.5)
	Zirconia	no defect	31.7 (4.3)	6.3 (0.9)	5.0 (1.3)	36.6 (4.8)	8.8 (1.0)	6.9 (0.3)
		3-wall defect	28.6 (4.5)	5.4 (0.8)	2.5 (1.8)	37.4 (3.7)	8.1 (1.0)	6.0 (0.7)
		1-wall defect	28.1 (3.6)	3.6 (2.0)	1.4 (1.1)	32.3 (3.3)	6.4 (0.8)	7.1 (0.9)
		Circular defect	17.9 (2.5)	4.4 (1.2)	0.6 (0.7)	24.2 (5.8)	5.8 (0.8)	1.5 (0.8)
	Bone level	no defect	30.7 (4.5)	2.7 (1.8)	1.7 (1.5)	41.4 (4.5)	7.0 (0.8)	5.4 (0.7)
		3-wall defect	21.8 (5.3)	2.0 (0.0)	1.6 (1.8)	29.4 (5.1)	5.0 (1.2)	1.6 (1.0)
		1-wall defect	25.3 (5.7)	5.0 (0.7)	1.7 (1.7)	28.6 (3.7)	6.2 (0.8)	3.3 (1.5)
		Circular defect	8.6 (1.8)	1.0 (1.1)	0.3 (0.7)	16.9 (3.6)	3.6 (1.8)	1.1 (0.9)
	BLX	no defect				27.3 (4.4)	3.9 (1.5)	0.9 (1.0)
		3-wall defect				20.3 (3.7)	0.5 (1.6)	0.6 (0.5)
		1-wall defect				20.8 (2.9)	1.5 (2.4)	0.5 (0.7)
		Circular defect				11.1 (2.5)	0.8 (1.6)	0.0 (0.0)

**Table 2 materials-13-04349-t002:** Inter- and intra-implant statistical analysis of the ISQ values.

*p*-Values (with Thread Cutting)	Block Density															
	D1				D2				D3				D4			
Statistical Analysis of ISQ Values	Tapered Effect	Tissue Level	Zirconia	Bone Level	Tapered Effect	Tissue Level	Zirconia	Bone Level	Tapered Effect	Tissue Level	Zirconia	Bone Level	Tapered Effect	Tissue Level	Zirconia	Bone Level
No defect vs. 3-wall defect	0.9487	0.9992	0.9598	0.9935	0.9465	0.9269	0.4023	0.7791	0.8801	0.8869	0.1027	0.4707	0.9782	0.4390	0.3070	0.0091
No defect vs. 1-wall defect	0.0593	0.4803	0.9389	0.9988	0.2165	>0.9999	0.5393	0.0032	0.0950	0.8930	0.0785	0.4634	0.0008	0.9983	0.2675	0.2013
No defect vs. circular defect	<0.0001	<0.0001	<0.0001	0.0084	0.0091	0.0004	<0.0001	0.0006	<0.0001	0.0020	0.0191	0.0025	<0.0001	0.0087	0.0002	<0.0001
3-wall defect vs. 1-wall defect	0.0320	0.3744	>0.9999	>0.9999	0.2500	0.6505	0.0115	0.0034	0.1219	0.4591	0.7565	0.9870	0.0096	0.5673	0.9836	0.1446
3-wall defect vs. circular defect	<0.0001	<0.0001	0.0005	0.0019	0.0047	<0.0001	<0.0001	0.0003	<0.0001	0.0024	0.0026	<0.0001	<0.0001	0.1916	0.0001	0.0002
1-wall defect vs. circular defect	0.0003	0.0001	0.0001	0.0548	0.0184	0.0001	<0.0001	<0.0001	<0.0001	0.0009	0.0008	0.0002	<0.0001	0.0396	<0.0001	0.0001
	No defect	3-wall defect	1-wall defect	circular defect	No defect	3-wall defect	1-wall defect	circular defect	No defect	3-wall defect	1-wall defect	circular defect	No defect	3-wall defect	1-wall defect	circular defect
Tapered effect vs. Tissue level	0.0001	<0.0001	0.0018	>0.9999	0.9984	0.5023	0.7124	0.0061	0.0009	0.0038	0.2347	0.0430	0.0001	<0.0001	0.0004	0.0006
Tapered effect vs. Zirconia	<0.0001	<0.0001	<0.0001	0.0221	<0.0001	<0.0001	<0.0001	0.9982	0.0100	0.9998	0.1205	0.0342	0.0410	0.0197	0.0671	0.9240
Tapered effect vs. Bone level	<0.0001	<0.0001	<0.0001	<0.0001	<0.0001	<0.0001	<0.0001	0.0149	0.6010	0.6515	0.2585	0.7754	0.0609	0.0004	0.0287	0.0003
Tissue level vs. Zirconia	0.2012	0.4833	0.0026	0.0221	0.0030	<0.0001	0.0004	0.0018	0.0472	0.0125	0.0350	0.0003	0.0021	0.0016	0.0035	0.0032
Tissue level vs. Bone level	0.2012	0.3644	0.0009	<0.0001	0.0076	<0.0001	<0.0001	<0.0001	0.0137	0.0033	0.0275	0.0597	0.0081	0.0842	0.0934	0.2921
Zirconia vs. Bone level	>0.9999	0.9971	0.9916	<0.000	0.4831	0.0092	0.0150	0.0037	0.6609	0.8336	0.9998	0.1892	0.9162	0.0438	0.5015	<0.0001
**Statistical analysis of** **torque in values**	Tapered effect	Tissue level	Zirconia	Bone level	Tapered effect	Tissue level	Zirconia	Bone level	Tapered effect	Tissue level	Zirconia	Bone level	Tapered effect	Tissue level	Zirconia	Bone level
No defect vs. 3-wall defect	0.1131	0.0946	<0.0001	0.0001	0.0036	0.0012	0.3156	0.8213	<0.0001	0.9556	0.8964	0.8686	0.6689	0.7050	0.1513	0.6360
No defect vs. 1-wall defect	0.1290	0.1378	<0.0001	0.7941	0.0006	<0.0001	0.0102	0.0538	<0.0001	0.9783	0.9968	0.0047	0.0739	0.9070	0.0167	0.0233
No defect vs. circular defect	0.0020	0.7579	0.3295	0.0003	0.7281	0.0723	0.0117	0.8542	<0.0001	0.2109	0.0022	0.3210	0.3482	0.0471	0.0033	0.1293
3-wall defect vs. 1-wall defect	0.7073	0.7766	0.0119	0.0432	0.7670	<0.0001	0.0421	0.0067	0.7207	0.7996	0.8186	0.0155	0.0219	0.9975	0.0412	<0.0001
3-wall defect vs. circular defect	0.7404	0.1172	0.0001	0.2238	0.0015	0.1331	0.0270	0.3107	<0.0001	0.5809	0.0054	0.9256	0.7533	0.6479	0.0889	0.0601
1-wall defect vs. circular defect	0.0808	0.1458	<0.0001	0.0236	0.0012	<0.0001	0.0053	0.2289	<0.0001	0.0006	0.0006	0.0001	0.0378	0.4954	0.6850	<0.0001
	No defect	3-wall defect	1-wall defect	circular defect	No defect	3-wall defect	1-wall defect	circular defect	No defect	3-wall defect	1-wall defect	circular defect	No defect	3-wall defect	1-wall defect	circular defect
Tapered effect vs. Tissue level	<0.0001	0.0002	0.0003	<0.0001	<0.0001	<0.0001	0.0014	<0.0001	0.1417	0.0014	<0.0001	0.0601	0.0001	0.0002	0.0010	0.0030
Tapered effect vs. Zirconia	<0.0001	0.7026	0.9931	0.0025	0.5478	0.0035	0.2119	0.0125	0.3608	0.0082	<0.0001	0.4118	0.0394	0.0008	0.0019	0.0069
Tapered effect vs. Bone level	0.0222	0.0003	0.1089	0.0013	0.2796	<0.0001	0.0037	0.5122	0.0030	0.0001	<0.0001	0.8711	0.0001	<0.0001	0.0001	<0.0001
Tissue level vs. Zirconia	0.0075	0.0002	0.0003	0.0010	<0.0001	<0.0001	0.1871	0.0137	0.0260	0.0446	<0.0001	0.0163	0.0007	0.0089	0.2505	0.3664
Tissue level vs. Bone level	<0.0001	0.7438	0.0984	<0.0001	0.0003	0.0111	0.7404	0.0002	0.9888	0.9947	0.0374	0.5853	0.3033	>0.9999	0.0196	0.0373
Zirconia vs. Bone level	0.0058	0.0001	0.0822	0.9942	0.5532	0.0015	0.4845	0.2055	0.0057	0.0006	0.6793	0.3098	0.0009	<0.0001	0.1779	0.0005

**Table 3 materials-13-04349-t003:** Inter- and intra-implant statistical analysis of torque-in values.

*p* Values(without Thread Cutting)	Block Density
D1		D2		D3		D4	
Statistical Analysis of ISQ Values	Tapered Effect	Tissue Level	Zirconia	Bone Level	BLX	Tapered Effect	Tissue Level	Zirconia	Bone Level	BLX	Tapered Effect	Tissue Level	Zirconia	Bone Level	BLX	Tapered Effect	Tissue Level	Zirconia	Bone Level	BLX
No defect vs. 3-wall defect	0.3387	0.9775	0.8464	0.4417	0.7500	0.9987	0.9997	0.4654	0.5492	0.6482	0.8811	0.2633	0.8812	0.1904	0.2622	0.2558	0.9974	0.9818	0.0003	0.0242
No defect vs. 1-wall defect	0.3351	0.5516	0.3889	0.0073	0.3386	0.0766	0.9770	0.0153	0.1285	0.6545	0.0199	0.3828	0.7801	0.0602	<0.0001	0.0332	0.3379	0.1511	0.0001	0.0328
No defect vs. circular defect	0.0047	0.0191	0.0034	0.0099	0.0001	<0.0001	0.0005	0.0077	0.0003	<0.0001	<0.0001	0.0283	<0.0001	<0.0001	<0.0001	<0.0001	0.2540	0.0035	<0.0001	<0.0001
3-wall defect vs. 1-wall defect	0.0003	0.0781	0.4268	0.5591	0.9755	0.0308	0.9788	0.9778	0.5764	0.0465	0.0803	0.9785	0.8520	0.9885	0.0144	0.0004	0.5076	0.0457	0.9833	0.9798
3-wall defect vs. circular defect	0.0017	0.0016	0.0018	0.0006	<0.0001	0.0003	0.0012	0.0016	0.0002	<0.0001	<0.0001	0.0079	0.0004	0.0014	<0.0001	<0.0001	0.0246	0.0007	0.0005	0.0007
1-wall defect vs. circular defect	0.0168	0.0002	<0.0001	0.0003	<0.0001	0.0014	0.0020	0.0002	0.0002	<0.0001	<0.0001	0.0062	0.0004	0.0003	<0.0001	<0.0001	0.0006	0.0338	<0.0001	0.0002
	No defect	3-wall defect	1-wall defect	circular defect		No defect	3-wall defect	1-wall defect	circular defect		No defect	3-wall defect	1-wall defect	circular defect		No defect	3-wall defect	1-wall defect	circular defect	
Tapered effect vs. Tissue level	0.0004	0.0001	<0.0001	0.5537		0.0032	0.0052	0.0004	0.3377		0.0003	0.0004	0.0086	0.9802		0.0002	<0.0001	0.0003	0.9988	
Tapered effect vs. Zirconia	0.0790	0.0976	0.0003	0.9339		0.7823	0.0286	<0.0001	0.3031		0.2588	0.9955	0.0182	0.0021		0.9994	0.9866	0.9904	0.2197	
Tapered effect vs. Bone level	<0.0001	<0.0001	<0.0001	0.0017		<0.0001	<0.0001	<0.0001	0.0014		<0.0001	0.6752	0.0578	0.5208		0.0653	0.0046	0.0404	0.8670	
Tapered effect vs. BLX	<0.0001	<0.0001	<0.0001	0.0102		0.0004	0.0014	<0.0001	0.4273		0.2189	0.4479	0.0012	0.9879		0.0007	<0.0001	<0.0001	0.0069	
Tissue level vs. Zirconia	0.0544	0.5475	0.3229	0.3580		<0.0001	0.1391	0.3373	0.9851		0.0002	0.0007	0.0022	0.0001		0.0020	0.0010	0.1347	0.1427	
Tissue level vs. Bone level	>0.9999	0.3265	0.3502	0.2188		0.0196	0.0049	<0.0001	0.0669		<0.0001	0.0005	0.0005	0.4837		0.0008	0.2091	0.9520	0.5139	
Tissue level vs. BLX	0.9539	0.8573	0.0250	0.8626		0.4586	>0.9999	0.2011	0.6643		0.0002	0.0097	0.1065	>0.9999		0.6327	0.1159	0.0092	0.0012	
Zirconia vs. Bone level	0.0814	0.0437	0.0812	0.0033		<0.0001	<0.0001	<0.0001	0.0010		0.0004	0.6233	0.3851	0.2777		0.2822	0.0080	0.1773	0.0092	
Zirconia vs. BLX	0.0118	0.2155	>0.9999	0.0221		0.0064	0.2658	0.8947	0.8753		>0.9999	0.3670	<0.0001	0.0001		0.0003	<0.0001	0.0003	0.0023	
Bone level vs. BLX	0.9388	0.3057	0.0025	0.0087		<0.0001	0.0002	<0.0001	0.0026		0.0002	0.4057	0.0008	0.4588		0.0002	0.0095	0.0010	0.0535	

**Statistical analysis of** **torque in values**	Tapered effect	Tissue level	Zirconia	Bone level	BLX	Tapered effect	Tissue level	Zirconia	Bone level	BLX	Tapered effect	Tissue level	Zirconia	Bone level	BLX	Tapered effect	Tissue level	Zirconia	Bone level	BLX
No defect vs. 3-wall defect	0.1539	0.9885	0.3582	<0.0001	0.0167	0.0433	0.2306	0.8411	<0.0001	0.9244	0.2614	0.1816	0.3881	0.1733	0.0631	0.0032	0.0003	0.5227	0.0127	0.0036
No defect vs. 1-wall defect	>0.9999	0.8026	0.7533	0.1956	0.5601	0.0022	<0.0001	0.0762	0.0437	0.0162	0.1591	<0.0001	0.0017	0.6875	<0.0001	0.0039	<0.0001	0.0043	0.0440	0.0668
No defect vs. circular defect	0.0291	0.0889	<0.0001	0.0003	0.0022	0.4924	0.1868	0.0050	<0.0001	0.0003	0.0004	0.0199	<0.0001	0.0001	<0.0001	<0.0001	0.0009	0.0003	0.0036	0.0127
3-wall defect vs. 1-wall defect	0.1539	0.9029	0.6602	0.0005	0.0835	0.0870	0.0008	0.0021	<0.0001	0.0026	0.9062	<0.0001	0.0004	0.0248	0.1363	0.7109	0.9194	0.0030	0.0717	0.7533
3-wall defect vs. circular defect	0.9991	0.0494	0.0002	0.3016	0.1417	0.0001	0.8544	0.0014	0.0343	0.0002	0.0006	0.1513	<0.0001	0.0046	0.0113	0.1435	0.9927	0.0007	0.2745	0.9600
1-wall defect vs. circular defect	0.0291	0.0078	<0.0001	0.0028	0.0090	<0.0001	<0.0001	<0.0001	<0.0001	0.0408	<0.0001	0.0002	<0.0001	<0.0001	0.0163	0.3057	0.9820	0.1816	0.0161	0.8947
	No defect	3-wall defect	1-wall defect	circular defect		No defect	3-wall defect	1-wall defect	circular defect		No defect	3-wall defect	1-wall defect	circular defect		No defect	3-wall defect	1-wall defect	circular defect	
Tapered effect vs. Tissue level	0.0004	0.0105	0.0004	<0.0001		0.0002	0.0001	0.0009	<0.0001		<0.0001	<0.0001	0.0014	<0.0001		<0.0001	0.0046	0.0011	0.0204	
Tapered effect vs. Zirconia	>0.9999	0.9030	0.8491	0.0003		>0.9999	0.0024	0.1058	0.0335		0.9954	0.9888	0.0009	0.1075		0.0108	0.3772	0.0006	0.0082	
Tapered effect vs. Bone level	>0.9999	0.0002	0.2624	0.0021		0.0067	<0.0001	0.0029	0.0024		0.0003	<0.0001	0.0006	0.0006		0.0008	<0.0001	0.0012	0.0034	
Tapered effect vs. BLX	0.0002	<0.0001	0.0007	<0.0001		0.0004	<0.0001	<0.0001	<0.0001		<0.0001	<0.0001	<0.0001	<0.0001		<0.0001	<0.0001	0.0002	<0.0001	
Tissue level vs. Zirconia	0.0004	0.0008	0.0003	0.1723		<0.0001	0.0026	0.1792	0.7451		<0.0001	<0.0001	0.2085	<0.0001		<0.0001	0.0160	0.1866	0.5668	
Tissue level vs. Bone level	0.0004	0.6647	0.0040	0.2661		<0.0001	0.2904	0.6786	0.7311		<0.0001	<0.0001	0.9681	0.7686		<0.0001	0.9917	0.2897	0.5368	
Tissue level vs. BLX	0.2483	0.8881	0.9380	0.8795		0.4958	0.6841	<0.0001	<0.0001		0.0014	0.3283	<0.0001	0.0014		0.0006	0.0262	0.0139	0.0051	
Zirconia vs. Bone level	>0.9999	0.0003	0.4451	0.9997		0.0040	0.0004	0.8596	>0.9999		<0.0001	<0.0001	0.0596	<0.0001		0.0180	0.0021	0.9837	0.0975	
Zirconia vs. BLX	0.0002	<0.0001	0.0009	0.2931		<0.0001	<0.0001	<0.0001	0.0025		<0.0001	<0.0001	<0.0001	<0.0001		0.0009	<0.0001	0.0015	<0.0001	
Bone level vs. BLX	0.0002	0.0019	0.0438	0.4541		<0.0001	0.0031	<0.0001	<0.0001		<0.0001	0.0037	<0.0001	0.0175		0.0006	0.0003	0.0040	0.1025

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
