# Peer review of "Relationship between Implant Geometry and Primary Stability in Different Bony Defects and Variant Bone Densities: An In Vitro Study"

_materials, 2020, doi:10.3390/ma13194349_

Round 1

Reviewer 1 Report

  • Abstract: please revise "the effect" as "the effects" in page 1 line15
  • Abstract: please revise "Spearman rho" as "Sperman's rho" in page 1 line 27 and 28.
  • Abstract: page 1 line 28, please rephrase the sentence "A slight increase in primary implant stability in all bone densities was achieved by omitting the thread cut."  It is difficult to understand. 
  • Introduction: please add the references (Modified surgical drilling protocols influence osseointegration performance and predict value of implant stability parameters during implant healing process. Huang et al. Clin Oral Investig. 2020 Jan 27.) after the sentence "drilling protocols, such as an underprepared drilling diameter in Page 2 line 44; and another reference (Evaluation of the implant stability and the marginal bone level changes during the first three months of dental implant healing process: A prospective clinical study. Su et al. Journal of the Mechanical Behavior of Biomedical Materials. Oct 2020, 110, 103899.) after the sentence"On the other hand, the implant's micro- and macromorphologies are also related to stability" in Page 2 line 46.
  • Results: in figures and tables: why no results were found in the BLX group.
  • Discussion: in page 4 line 174, why the non-self-cutting threads can obtain higher primary implant stability than self-cutting threads? Please extend the explanation.
  • Discussion: in page 4 line 181-191, please extend the explanation regarding the lower values were found in torque-in than the torque-out.
  • Discussion: please extend the description regarding tapered effect is beneficial in D4 bone density and circular effect.

Author Response

Dear Editors, dear reviewers,

Thank you very much for the opportunity of the major revision and the helpful comments on the manuscript “Relationship between implant geometry and primary stability in different bony defects and variant bone densities: an in vitro study " (materials-941202). Below you will find a checklist regarding the comments and your desired changes point by point. All notes and comments were worked in new version and were highlighted with “track change”.

Reviewer 1:

Concerns of reviewer 1:

Abstract: please revise "the effect" as "the effects" in page 1 line15, Abstract: please revise "Spearman rho" as "Spearman's rho" in page 1 line 27 and 28.

Response to reviewer: Thank you very much for your comments. We corrected these points.

 Concerns of reviewer 1:

Abstract: page 1 line 28, please rephrase the sentence "A slight increase in primary implant stability in all bone densities was achieved by omitting the thread cut."  It is difficult to understand. 

Response to reviewer: This was rephrased.

Text change: “The implant insertion without any thread cut led to a small improvement in primary implant stability in all bone densities.”

 Concerns of reviewer 1:

Introduction: please add the references (Modified surgical drilling protocols influence osseointegration performance and predict value of implant stability parameters during implant healing process. Huang et al. Clin Oral Investig. 2020 Jan 27.) after the sentence "drilling protocols, such as an underprepared drilling diameter in Page 2 line 44; and another reference (Evaluation of the implant stability and the marginal bone level changes during the first three months of dental implant healing process: A prospective clinical study. Su et al. Journal of the Mechanical Behavior of Biomedical Materials. Oct 2020, 110, 103899.) after the sentence"On the other hand, the implant's micro- and macromorphologies are also related to stability" in Page 2 line 46.

Response to reviewer: Thank you for the helpful references. These references were added

 Concerns of reviewer 1:

Results: in figures and tables: why no results were found in the BLX group.

Response to reviewer: Figures and tables were divided into subgroups of implant insertions “with” and “without” implant thread cut. As BLX implants have a special self-cutting thread design, BLX implants were only inserted without thread cut. The company's drilling protocol does not include a tap. Therefore, results were only shown in figure 4 and all tables in the subgroup “no thread cut”.

Please also see methods section “Except for the BLX implants with a self-cutting thread design, all the other implant types were inserted one time with and without thread cutting.”

 Concerns of reviewer 1:

Discussion: in page 4 line 174, why the non-self-cutting threads can obtain higher primary implant stability than self-cutting threads? Please extend the explanation.

Response to reviewer: Thank you very much for your comments. We added an explanation.

Text change: “Threads contribute to primary stability by increasing the initial contact with the underlying bone [24]. Self-tapping implants are usually designed to avoid the use of thread-cutting techniques to prepare the implant site, which is replaced by the action of cutting edges integrated into the lower apical part of the implant. Threads, however, differ in the way they transmit loads to adjacent bone [24]. V-shaped threads and implants with reverse buttress-threaded implants showed to transmit axial force through a combination of compressive, tensile, and shear forces. Furthermore, as the thread face angle increases, the shear forces generated by the various thread shapes increase, too [25]. Non-self-cutting blades create a lateral compression with an increased contact surface area, thus improving primary stability.

Concerns of reviewer 1:

Discussion: in page 4 line 181-191, please extend the explanation regarding the lower values were found in torque-in than the torque-out.

Response to reviewer: We added an explanation.

Text change: “The torque out values were lower when compared to the insertion torque. This is due to the correlation of the torque out value with the gripping volume, whereas the insertion torque value is rather associated with individual bone compression, e.g. due to different bone densities (Ting e al.).”

Concerns of reviewer 1:

Discussion: please extend the description regarding tapered effect is beneficial in D4 bone density and circular effect.

Response to reviewer: We improved the explanation.

Text change: “Moreover, compared with dental implants with parallel walls, the tapered shape may lead to better primary stability due to an improved compressive forces distribution. However, several studies have reported controversial results with lower stability in tapered implants than in parallel ones [27, 28], whereas other studies have found the opposite [29, 30]. In our study tapered effect might have positively influenced implant stability especially in D4 bone density and circular effect due to the improved compressive forces at the implant tip.”

Thank you for your support,

Kind regards,

Kristian Kniha, DDS, Priv. Doz.

Reviewer 2 Report

The topic of this article entitled “Relationship between implant geometry and primary stability in different bony defects and variant bone densities: an in vitro study” is aimed to assess the interactions among specific variables, like implant geometry and stability. The effect of bony condition was also studied.

It is an interesting topic and within the journal's scope. Nevertheless, this reviewer would suggest some improvements, before further considerations.  The study has certainly new information with interesting aspects on dental sciences. This article also aims to translate its main key-concepts to applied medicine in the very next future.

Introduction: Authors have reported several important topics related to bone tissue healing and repairing. However, poor has been reported on the role of resident stem cells, which can act as the immunomodulatory and pro-osteogenic activities in the local environment (Please, see and discuss: Ballini et al. Mesenchymal Stem Cells as Promoters, Enhancers, and Playmakers of the Translational Regenerative Medicine 2018. Stem Cells Int. 2018 Oct 30;2018:69274019 – and -  Ballini et al. Mesenchymal stem cells as promoters, enhancers, and playmakers of the translational regenerative medicine. Stem Cells Int. 2017:3292810).

Authors have consistently discussed on the role of tissue engineering. However, something more should be discussed about the role of specific “biomaterials” or “scaffolds” as study model. In this light it’s important to briefly describe something about the “safe” in-vitro reparative models, working without any additive (e.g. BSA) to apply safely on humans (Please, see and discuss “Highly Efficient In Vitro Reparative Behaviour of Dental Pulp Stem Cells Cultured with Standardised Platelet Lysate Supplementation. Stem Cells Int. 2016;2016:7230987.”) On the other hand, the strategies to chose biomaterials and scaffolds, and the study on the managing of loads, should be briefly described in the discussion section, highlighting the role of pre-clinical investigations on this matter (Please, see and discuss “Marrelli M, Maletta C, Inchingolo F, Alfano M, Tatullo M. Three-point bending tests of zirconia core/veneer ceramics for dental restorations. Int J Dent 2013; 2013, 831976.”).

- Conclusions should be improved with clear take-home messages.

Minor suggestion:

- In the whole text there are some typos here and there: authors should carefully revise the text before resubmission.

Author Response

Dear Editors, dear reviewers,

Thank you very much for the opportunity of the major revision and the helpful comments on the manuscript “Relationship between implant geometry and primary stability in different bony defects and variant bone densities: an in vitro study " (materials-941202). Below you will find a checklist regarding the comments and your desired changes point by point. All notes and comments were worked in new version and were highlighted with “track change”.

Reviewer 2:

The topic of this article entitled “Relationship between implant geometry and primary stability in different bony defects and variant bone densities: an in vitro study” is aimed to assess the interactions among specific variables, like implant geometry and stability. The effect of bony condition was also studied.

It is an interesting topic and within the journal's scope. Nevertheless, this reviewer would suggest some improvements, before further considerations.  The study has certainly new information with interesting aspects on dental sciences. This article also aims to translate its main key-concepts to applied medicine in the very next future.

Introduction: Authors have reported several important topics related to bone tissue healing and repairing. However, poor has been reported on the role of resident stem cells, which can act as the immunomodulatory and pro-osteogenic activities in the local environment (Please, see and discuss: Ballini et al. Mesenchymal Stem Cells as Promoters, Enhancers, and Playmakers of the Translational Regenerative Medicine 2018. Stem Cells Int. 2018 Oct 30;2018:69274019 – and -  Ballini et al. Mesenchymal stem cells as promoters, enhancers, and playmakers of the translational regenerative medicine. Stem Cells Int. 2017:3292810). Authors have consistently discussed on the role of tissue engineering. However, something more should be discussed about the role of specific “biomaterials” or “scaffolds” as study model. In this light it’s important to briefly describe something about the “safe” in-vitro reparative models, working without any additive (e.g. BSA) to apply safely on humans (Please, see and discuss “Highly Efficient In Vitro Reparative Behaviour of Dental Pulp Stem Cells Cultured with Standardised Platelet Lysate Supplementation. Stem Cells Int. 2016;2016:7230987.”) On the other hand, the strategies to chose biomaterials and scaffolds, and the study on the managing of loads, should be briefly described in the discussion section, highlighting the role of pre-clinical investigations on this matter (Please, see and discuss “Marrelli M, Maletta C, Inchingolo F, Alfano M, Tatullo M. Three-point bending tests of zirconia core/veneer ceramics for dental restorations. Int J Dent 2013; 2013, 831976.”).

- Conclusions should be improved with clear take-home messages.

Minor suggestion:

- In the whole text there are some typos here and there: authors should carefully revise the text before resubmission.

Response to reviewer: Thank you for your efforts and the comments. The authors politely ask if there is any mix-up regarding the comments of reviewer 2 regarding our manuscript, as the mentioned references (e.g. the clinical process of “Mesenchymal Stem Cells as Promoters, Enhancers, and Playmakers of the Translational Regenerative”) don't really fit and seem a little off topic when compared to this in vitro study about primary stability of implants inserted into different blocks. Furthermore, “Authors have reported several important topics related to bone tissue healing and repairing.” does not completely fit to our version of this in vitro study. Same for “Authors have consistently discussed on the role of tissue engineering.”, as we have not discussed tissue engineering in the first version. We apologize for the confusion and would of course appreciate any helpful comments on our manuscript.

Thank you for your support,

Kind regards,

Kristian Kniha, DDS, Priv. Doz.

Round 2

Reviewer 2 Report

The topic of this article entitled “Relationship between implant geometry and primary stability in different bony defects and variant bone densities: an in vitro study” is aimed to assess the interactions among specific variables, like implant geometry and stability. The effect of bony condition was also studied.

It is an interesting topic and within the journal's scope. Nevertheless, this reviewer would suggest some improvements, before further considerations.  The study has certainly new information with interesting aspects on dental sciences. This article also aims to translate its main key-concepts to applied medicine in the very next future.

1. Introduction: Authors have reported several important topics related to bone tissue healing and repairing (as an example: "Primary stability may be affected by many factors. Besides clinical factors such as resident stem cells, which can act as immunomodulatory and pro-osteogenic activities in the local environment, material-related factors also play a key-role").

However, poor has been reported on the role of resident/oral-derived/bone-derived stem cells, which can act as the immunomodulatory and pro-osteogenic activities in the local (bone/oral) environment (Please, see and discuss: Ballini et al. Mesenchymal Stem Cells as Promoters, Enhancers, and Playmakers of the Translational Regenerative Medicine 2018. Stem Cells Int. 2018 Oct 30;2018:69274019 – and -  Ballini et al. Mesenchymal stem cells as promoters, enhancers, and playmakers of the translational regenerative medicine. Stem Cells Int. 2017:3292810).

2. Something more should be discussed about the role of specific “biomaterials” or “scaffolds” as study model for bone defect replacement. In this light it’s important to briefly describe something about the strategies to chose biomaterials and scaffolds, and the study on the managing of loads, should be briefly described in the discussion section, highlighting the role of pre-clinical investigations on this matter (Please, see and discuss “Marrelli M, Maletta C, Inchingolo F, Alfano M, Tatullo M. Three-point bending tests of zirconia core/veneer ceramics for dental restorations. Int J Dent 2013; 2013, 831976.”).

3. Conclusions should be improved with clear take-home messages.

Minor suggestion:

4. In the whole text there are some typos here and there: authors should carefully revise the text before resubmission.

Author Response

Dear Editors, dear reviewers,

Thank you very much for the opportunity of the major revision and the helpful comments on the manuscript “Relationship between implant geometry and primary stability in different bony defects and variant bone densities: an in vitro study " (materials-941202). Below you will find a checklist regarding the comments and your desired changes point by point. All notes and comments were worked in new version and were highlighted with “track change”.

Reviewer 2:

The topic of this article entitled “Relationship between implant geometry and primary stability in different bony defects and variant bone densities: an in vitro study” is aimed to assess the interactions among specific variables, like implant geometry and stability. The effect of bony condition was also studied.

It is an interesting topic and within the journal's scope. Nevertheless, this reviewer would suggest some improvements, before further considerations.  The study has certainly new information with interesting aspects on dental sciences. This article also aims to translate its main key-concepts to applied medicine in the very next future.

Response to reviewer: Thank you for your comments. We definitely agree with the reviewer that “Oral Stem Cells in Tissue Engineering and Regenerative Medicine” is a very interesting topic and it should be reported.

Concerns of reviewer 2:

  1. Introduction: Authors have reported several important topics related to bone tissue healing and repairing (as an example: "Primary stability may be affected by many factors. Besides clinical factors such as resident stem cells, which can act as immunomodulatory and pro-osteogenic activities in the local environment, material-related factors also play a key-role").

However, poor has been reported on the role of resident/oral-derived/bone-derived stem cells, which can act as the immunomodulatory and pro-osteogenic activities in the local (bone/oral) environment (Please, see and discuss: Ballini et al. Mesenchymal Stem Cells as Promoters, Enhancers, and Playmakers of the Translational Regenerative Medicine 2018. Stem Cells Int. 2018 Oct 30;2018:69274019 – and -  Ballini et al. Mesenchymal stem cells as promoters, enhancers, and playmakers of the translational regenerative medicine. Stem Cells Int. 2017:3292810).

Response to reviewer: We added the references in the introduction.

Additionally, the role of immunomodulatory and pro-osteogenic activities was discussed in the discussion.

Text change: “Adequate initial stability is required to allow the implant to withstand micromovements before osseointegration is achieved. Subsequently, primary stability is one of several factors that is positively associated with secondary stability due to new bone formation [3]. Primary stability may be affected by many factors. Besides clinical factors such as resident stem cells, which can act as immunomodulatory and pro-osteogenic activities in the local environment, material-related factors such as implant design and drilling protocols play a key-role, too [4, 5]. On one hand, drilling protocols, such as an underprepared drilling diameter [6, 7] and …

“After this initial in vitro experiment presented here, clinical healing processes should also be

considered. For example, the clinical microenvironment of the wound is a key factor in the implant integration. In regenerative medicine dental mesenchymal stem cells are important in terms of their potential to differentiate into osteogenic lines around implants [4, 5]. They may have the future potential to improve bone healing, especially in relation to peri-implant bony defects.”

Concerns of reviewer 2:

  1. Something more should be discussed about the role of specific “biomaterials” or “scaffolds” as study model for bone defect replacement. In this light it’s important to briefly describe something about the strategies to choose biomaterials and scaffolds, and the study on the managing of loads, should be briefly described in the discussion section, highlighting the role of pre-clinical investigations on this matter (Please, see and discuss “Marrelli M, Maletta C, Inchingolo F, Alfano M, Tatullo M. Three-point bending tests of zirconia core/veneer ceramics for dental restorations. Int J Dent 2013; 2013, 831976.”).

Response to reviewer: This was better discussed and we added PRF as an important biomaterial for bone defects.

Text change: They may have the future potential to improve bone healing, especially in relation to peri-implant bony defects. Furthermore, biomaterials such as Platelet-Rich Fibrin could be used to fill the defects around implants between bone tissue and the implant surface in a clinical scenario [40].“

 Concerns of reviewer 2:

  1. Conclusions should be improved with clear take-home messages.

Minor suggestion:

Response to reviewer: Was improved

Text change: “In terms of primary stability, our results indicated that no significant loss of stability should be expected in smaller three-wall and one-wall peri-implant defects. However, a significant reduction in primary stability was observed especially in circular defects. As a counteract, a slight increase in primary stability could be achieved by omitting the thread cut bevor the implant placement. “

 Concerns of reviewer 2:

  1. In the whole text there are some typos here and there: authors should carefully revise the text before resubmission.

Response to reviewer: Thank you, grammar and language were checked using a professional proofread. Grammar and language were corrected throughout the manuscript.

Thank you for your support,

Kind regards,

Kristian Kniha, DDS, Priv. Doz.